# Credit Card Fraud Detection in Card-Not-Present Transactions: Where to Invest?

**Igor Mekterović** *[ID], **Mladen Karan, Damir Pintar** [ID] **and Ljiljana Brkić**

Faculty of Electrical Engineering and Computing, University of Zagreb, 10000 Zagreb, Croatia; mlade.karan@fer.hr (M.K.); damir.pintar@fer.hr (D.P.); ljiljana.brkic@fer.hr (L.B.)
* Correspondence: igor.mekterovic@fer.hr; Tel.: +385-1-61-29-790

**Abstract:** Online shopping, already on a steady rise, was propelled even further with the advent of the COVID-19 pandemic. Of course, credit cards are a dominant way of doing business online. The credit card fraud detection problem has become relevant more than ever as the losses due to fraud accumulate. Most research on this topic takes an isolated, focused view of the problem, typically concentrating on tuning the data mining models. We noticed a significant gap between the academic research findings and the rightfully conservative businesses, which are careful when adopting new, especially black-box, models. In this paper, we took a broader perspective and considered this problem from both the academic and the business angle: we detected challenges in the fraud detection problem such as feature engineering and unbalanced datasets and distinguished between more and less lucrative areas to invest in when upgrading fraud detection systems. Our findings are based on the real-world data of CNP (card not present) fraud transactions, which are a dominant type of fraud transactions. Data were provided by our industrial partner, an international card-processing company. We tested different data mining models and approaches to the outlined challenges and compared them to their existing production systems to trace a cost-effective fraud detection system upgrade path.

**Keywords:** credit card fraud detection; card-not-present; data mining; feature engineering

## 1. Introduction

According to Statista [1], global retail e-commerce will reach almost 7 k billion US dollars in 2023 (Figure 1). E-commerce heavily relies on credit cards as a means of payment and credit card adoption, and the number of credit card transactions grows accordingly. Sadly, it seems that fraudsters are keeping track and even thriving in this growing environment: the credit card fraud ratio is the same or perhaps slightly growing (note the green line in Figure 2). Though credit card fraud percentages are seemingly small (around 0.04%), the losses are staggering in absolute numbers. For instance, the annual value of losses on card-not-present(CNP) fraud for debit and credit cards issued in the United Kingdom (UK) alone for 2019 amounted to 470.2 million GBP [2].

Over the years, technology has significantly changed, and so have the fraud patterns. Today, CNP is a dominant type of fraud, as visible in Figure 2, and it is reported that [3]: "CNP fraud accounted for €1.43 billion in fraud losses in 2018 (an increase of 17.7% compared with 2017)." This share has been growing steadily since 2008 (not displayed in the chart). For that reason, in this paper, we focused on CNP transactions alone.

Our task was to consider how to efficiently enhance existing real-world credit card processing infrastructure using data mining techniques while at the same time considering the practical implications.

Credit card transactions must satisfy two conflicting properties from a technical standpoint: they must be fast (measured in milliseconds), and they must be secure. This is why credit card transaction processing is performed in multiple stages: real-time, near

real-time, and offline [4]. In real-time, only fundamental checks are performed (PIN, balance, ... ), followed by near real-time, where traditionally rule engines are used.

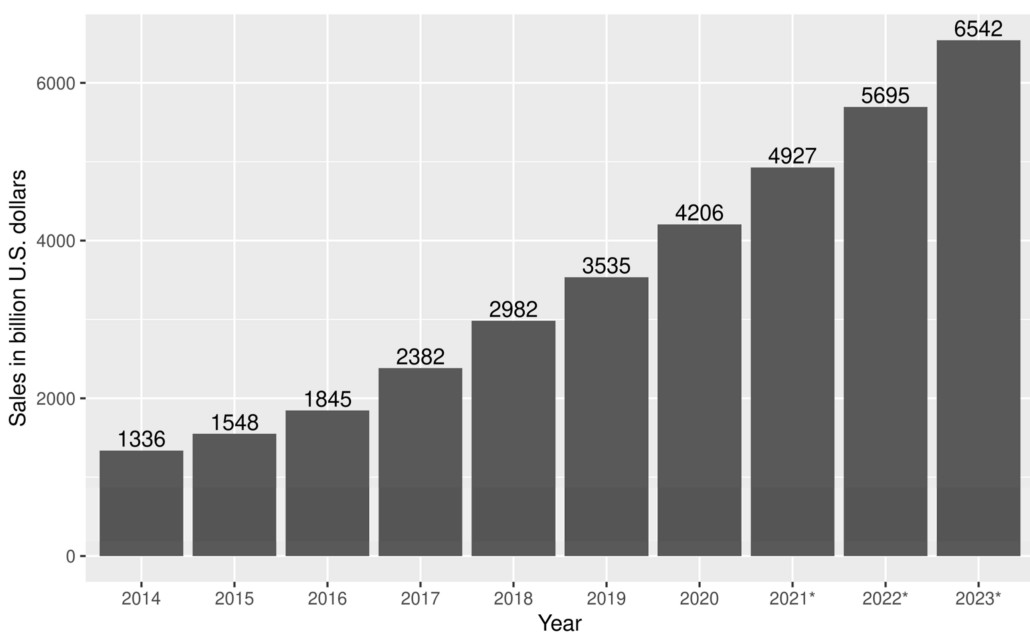

**Figure 1.** Retail e-commerce sales worldwide from 2014 to 2023 (in billion U.S. dollars).

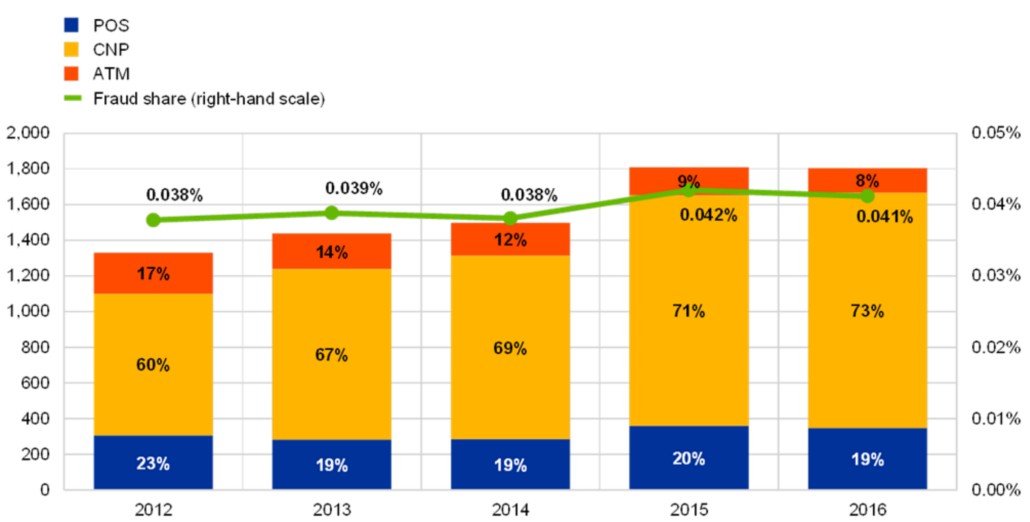

**Figure 2.** Evolution of the total value of card fraud using cards issued within Single Euro Payments Area.

Figure 3 shows simplified authorization and fraud detection flow.

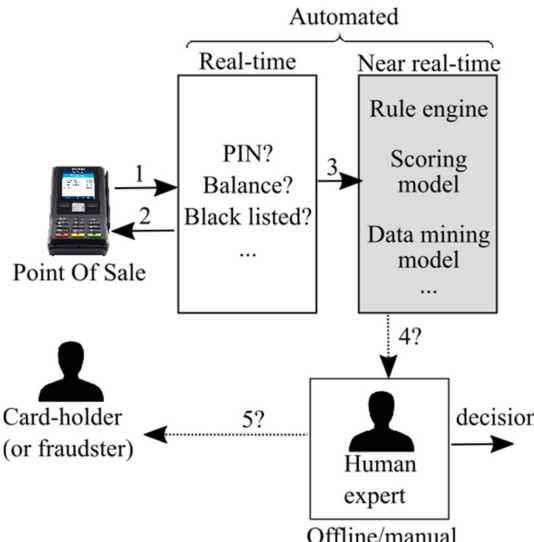

**Figure 3.** Simplified authorization and fraud detection flow [4].

Rule engines validate transactions against a set of manually defined rules and are probably still dominant in production systems (this information is hard to confirm as credit card processors understandably tend to be secretive about their inner workings). In recent years, rule engines are complemented with various machine learning models to boost the overall fraud detection precision. A significant number of papers have been published in the last ten years on the subject [4,5] such as [6–8]. However, it is hard to say to what extent this trend penetrated the production systems, which are very conservative and secretive. Rule engines have the lovely property of being interpretable, which cannot be said for most data mining models. Finally, the final verdict on the fraud case is given "offline" by a human expert having all the relevant information at his or her disposal. In this paper, using a real-world dataset in cooperation with our industrial partner, we address credit card fraud detection challenges to produce a proper business plan: where best to invest time and money considering the usual budgetary and time constraints. In other words, we perform triage of sorts, gaining insights that might generalize well to similar datasets and fraud detection problems. We contribute by studying this problem from the wider, business perspective (on how to build a fraud detection system efficiently and practically), from the architectural perspective (which relates to the scalability), and by corroborating findings in the literature pertaining to the algorithm selection and feature engineering.

## 2. Credit Card Fraud Detection Challenges

In a previous work [4], we performed a systematic review of data mining approaches to credit card fraud detection and identified the significant challenges in this area.

### 2.1. Lack of Data

Lack of data can be considered in two contexts: lack of literature on the topic and lack of training/test data (public credit card transactions databases). The latter is a problem for the scientist and not so much for the industry, as credit card processing houses have vast amounts of data. The former is often cited as a problem, but we respectfully disagree as there are many papers on the topic and even books (see [4]). It could be argued that there is the opposite problem—surveying and assimilating the voluminous and scattered literature to discern the best practices and methodologies.

### 2.2. Feature Engineering

Feature engineering is a classic topic in data mining and is particularly important in credit card fraud detection. Credit card processing firms and banks usually possess a rich set of features on credit cardholders that can be used to build a user/card profile, especially

when enriched with values aggregated from the card's previous transactions that sketch the card profile.

An interesting exception is systems in which the main payment instrument is prepaid cards that are not associated with a person. Prepaid cards are rarely topped up with money. The lifetime of a card is relatively short—from months to a year, for example. Therefore, there is a limited set of features at disposal and little information from which to create a card model. Such a system is described in [8] and in predicting card fraud, the authors used a dozen features as opposed to the few hundred that we used in our simulations.

### 2.3. Scalability

Scalability is a technical problem often ignored in the literature. One must strive to design robust and scalable systems to sustain a continual, large stream of transactions.

### 2.4. Unbalanced Class Sizes

As seen in Figure 2, the fraud ratio is minuscule—well below 0.1%. The class size imbalance problem is not exclusively present in fraud detection. It is common in many other areas including the detection of diseases in medical diagnosis [9,10], facial recognition [11,12], oil spill detection [13,14], earthquake prediction [15], email tagging [16], and the detection of potential customers in the insurance business [17]. Usual model quality measures (such as accuracy) are not suitable for these problems. Special care needs to be taken when evaluating the methods and measures. Most data mining algorithms are not designed to cope with such class disbalance. This issue can be addressed on an algorithmic level, but is typically addressed on the data level [18]. At the algorithmic level, algorithms themselves are adjusted to cope with the detection of the minority class, while at the data level, a pre-processing step is performed to rebalance the dataset. Various pre-processing techniques have been proposed to overcome the class imbalance problem on the data level including the dominant oversampling [19,20], undersampling [21,22], or a combination of both ensemble learning techniques [23] and cost-sensitive learning [24,25].

### 2.5. Concept Drift

Credit card fraud patterns change over time as the market and technology change, and both fraudsters and card processors adapt to the changes. This changes the underlying patterns and data and is referred to as "concept drift" [26].

Predictive models that operate in these settings need to have mechanisms to: (i) detect concept drift and adapt if needed; and (ii) distinguish drifts from noise and be adaptive to changes, but robust to noise. Simply put, models become stale and obsolete and must be refreshed or evolved.

Existing concept drift detection techniques are either statistically-based [27–29], window-based [30,31], or ensemble-based [32–34]. Readers interested in adapting predictive models to concept drifts (i.e., adaptive learning) are referred to in recent reviews [35,36] and the most cited review dealing with the concept drift [37].

### 2.6. Performance Measures

As often quoted, "That which cannot be measured cannot be improved", so it is essential to define a fitting metric for our models. There is a myriad of various metrics proposed in the literature [4], and in our work, we proposed a simple and informative chart to compare the competing models. Fraud detection is typically defined as a classification task: a transaction is classified as either fraud or non-fraud. In our opinion, it should be considered from the detection perspective: a set of transactions is ranked according to the probability of being fraudulent, which maps very well to the business case. Since transactions must ultimately be examined by the human expert, it is beneficial to rank them according to the fraud probability. One could define a "fraud threshold" at, for example, 50% probability. Still, it is irrelevant: a limited number of human experts in a limited amount of time will only be able to analyze a limited number of transactions, and

they should do it in the descending order of fraud probability. Credit card processors can trade-off the fraud loss with the analyst cost and achieve an optimal balance.

*2.7. Model Algorithm Selection*

Finally, many different data mining algorithms can address this problem. Each of them presents an optimization problem with many hyperparameters to tune. Furthermore, they can be combined to form ensembles and so forth. It is impossible to "try them all", so for practical reasons, the "best" algorithm or a shortlist of algorithms should be chosen in the first step to invest resources in.

## 3. Related Work

Various approaches have been proposed to solve the issue of detecting fraud in financial transactions. Traditional fraud prevention mechanisms in banks are mostly based on manpower-based rules such as the one presented in [38]. The rules describe the circumstances of a financial transaction that is considered suspicious and potentially worth checking. Rule-based solutions are flexible, but also challenging and time-consuming to implement and maintain as it requires the diligent definition of every single rule for some possible anomaly. If an expert fails to define a suspicious situation, undetected anomalies will happen, and nobody will be aware of them. On the other hand, as time passes by and credit card fraud patterns change, some rules become obsolete. This imposes the need to periodically assess the usefulness of a rule within a set of rules and decide whether to keep or drop the rule from the pool. The typical development of a fraud detection system starting with the application of a rule-based approach that initially worked well is described in [9]. The paper describes (which is consistent with our industry partners' experience) that as the ruleset increases, the effort to maintain a transaction monitoring system also increases, and consequently, the accuracy of fraud detection decreases. An interesting approach that assigns a normalized score to the individual rule, quantifying the rule influence on the pool's overall performance, is described in [39].

To improve detection and mitigate the limitations of rule-based systems, fraud detection solutions employ machine learning techniques divided into supervised and unsupervised [40]. In supervised techniques, models developed using annotated samples of fraudulent and non-fraudulent transactions are used to classify transactions as fraudulent or genuine. In contrast, unsupervised techniques seek those accounts, customers, transactions, etc., which raise doubts by differing significantly from most data. The recognition of such unalike data is called anomaly detection or outlier detection. A lot of outlier detection algorithms have been proposed in the literature, many of them being cluster-based [41–44]. Contrary to supervised approaches, unsupervised outlier detection does not require annotated transactions and can detect unforeseen outlying cases. The uncertainty factor in unsupervised techniques is that we do not have the annotated set to compare them with and are unsure about the results. Since banks have a vast number of high-quality fraud-labeled data records, supervised methods prevail. Neural networks have been used to detect fraud due to their huge popularity in the 1990s (e.g., [45,46]) and now again, with the advent of deep learning (e.g., [47,48]), but practically "all" machine learning algorithms were tested against this problem, with logistic regression [49–52], SVM [49–51,53,54], and random forests [49–52,55,56] being the most popular ones. Other supervised approaches include decision trees [53,56–59], Bayesian models [7,41,57,58,60], association rules [61], Hidden Markov model [8], etc.

The techniques mentioned differ in their computational complexity. Some, like neural networks and Bayesian networks, are intrinsically computationally intensive while, for example, for the K-nearest neighbor algorithm, computation time is usually very low [49]. Another critical aspect of the techniques used is the ability to support the well-known concept of drift problem. A supervised model must be periodically retrained to address concept drift. Robinson and Aria showed in [8] that most techniques do not directly address

concept drift. Unaddressed or poorly addressed concept drift leads to sporadic updates of the fraud detection model, resulting in periods of poor-quality fraud detection.

Ensemble (hybrid) models combining multiple distinct models have been the "gold standard" lately. It is generally accepted in the statistics and machine-learning community that the combination of different algorithms tends to produce superior results. Using hybrid models aims to make the most of each algorithm.

Digital payment platform PayPal, for example, has developed its artificial intelligence software to combat fraud. Their experience shows that the most effective approach in many cases is using an ensemble of decision trees, neural networks, and logistic regressions [62]. In LogSentinel [63], for example, unsupervised learning is combined with rule-based fraud detection to merge the power of machine learning with the adaptability of domain expertise. Another example of a hybrid approach for credit card fraud detection presented in [64] combines a rule-based filter, Dempster–Shafer adder, transaction history database, and Bayesian learner. The main advantage of ensemble models is their increased accuracy, but this comes at a raised computation cost and less intuitive or non-existent interpretation. However, none of the work we found deals with practical issues that we considered here, like cost-efficiency, scalability, maintenance, etc.

## 4. Experiment

This chapter describes the data, methodology, baseline system, and experiment results regarding the outlined challenges.

### 4.1. Baseline System

The baseline system was our industrial partner's production system—an international card processing company. It is a long-standing and well-functioning system that already includes a data mining model for scoring transactions. The task was to improve the system with a newly developed data mining model. In doing so, there are many issues to address (i.e., invest resources in). Our goal was to end up with a cost-effective solution that will provide the most considerable improvement in the fraud detection rate. The current baseline setup (Figure 4) comprises real-time checks (not interesting in this context) and two-stage near-real-time checks: SM (scoring model) and RE (rule engine) before potential fraud cases are forwarded to the analyst. SM is fed with transactional data and several aggregated variables, producing a fraud score—an integer in the [0, 400] range. SM was developed using the logistic regression module of SAS enterprise miner software. RE uses the same variables as the SM to add a newly computed fraud score, which may result in rules such as "if a fraud score is greater than 300 and country of origin is ABC and . . . ". If any of those custom rules is triggered, a "fraud case" is packaged and deployed to the analyst for the final verdict. RE was developed in-house with rules defined both on the general level and on the per-client level (e.g., a bank that is a client of the processing house can impose some private local rules). RE's apparent drawback is that with the continually rising number of rules and the segmentation according to the clients, the RE-based system can become hard to maintain.

In this data pipeline (Figure 4), the new model can be positioned in three places:

- Position A: parallel to the SM;
- Position B: parallel to RE, with the model being aware of the fraud score; and
- Position C: after the rule engine, with the model being aware of the rule count.

It is helpful to consider various placements as they impose different module coupling restrictions and overall transaction processing speed. For instance, should models A and C behave comparably, model A would be preferred as it is independent and can begin to process transactions sooner. In other words, with model placement evaluation, we are evaluating whether existing production SM and RE modules contain any additional knowledge that we cannot span with the newly developed model.

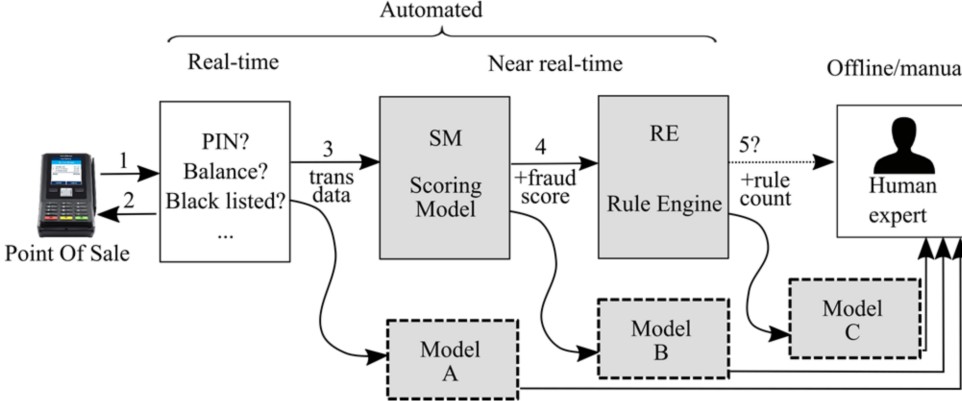

**Figure 4.** Diagram of the baseline (production) system and possible inclusion positions of a newly developed model.

*4.2. Dataset and Experiment Setup*

Our industrial partner's real-world dataset consisted of 197,471 transactions that took place over three months. Each row contains 377 features, of which:

- 66 are transaction features;
- 305 are aggregated features computed from the card's previous transactions. For instance, one variable is "the number of ATM transactions in the past 30 days". Current models use only eight aggregated features, and we have aggressively expanded the feature set to evaluate the aggregated features' impact on the model quality. It should be noted that aggregated features are not free of cost—they incur significant resource costs: to calculate them in near real-time, one must have transaction history at their disposal for fast computation of features. Keeping that in mind, the transaction rate of credit card processing can be quite a technological challenge;
- Depending on the model position, fraud_score and rule_count features are available to models B and C; and
- Target variable: whether a transaction is a fraud or not.
- For the model training purposes, the dataset was divided into training and test datasets:
- Training dataset: 70% of transactions, roughly first two months.
- Test dataset: 30% of transactions, roughly the third month.

We decided to divide the transactions chronologically (instead of sampling) to achieve a realistic scenario where the models need to predict future events based on historical data. The test dataset was used strictly for testing, and model scores in the testing phase were in no way used to improve the models (cross-validation on the training dataset was used for that purpose).

This dataset was already under-sampled to include all fraudulent transactions, having a fraud rate of 5%.

In our experiment, we varied the following properties:

- Model position: A, B, or C.
- Fraud percentage: 5% or 50%. The latter is obtained in two ways:
  - Undersampling of the majority class while preserving all fraud transactions— the resulting dataset has ~14 k transactions and is referred to as "small".
  - Combination of undersampling and oversampling—the resulting dataset has ~120 k transactions and is referred to as "balanced".
- Basic (transactional) set of 66 features and the full set of features. The former is here referred to as "trans".

Table 1 shows the abbreviations for the properties above-mentioned that are used to present results in the following text:

**Table 1.** Abbreviations used to present experiment results.

| Abbreviation | Meaning |
| --- | --- |
| trans | Only basic (transactional) set of 66 features. If omitted, full feature set, which includes aggregated features, is used. |
| 5 | 5% fraud rate |
| 50 | 50% fraud rate |
| {small ∣ balance} | small is obtained through undersampling of majority class and balanced with a combination of undersampling and oversampling. If none of these appears, then the integral dataset has been used. |
| A, B, C | Model position, see Figure 4 |
| RF, LR, or NN | random forest, logistic regression, or neural network, respectively |

For instance, label **RF.50.A.sm.trans** means: random forest model trained on the undersampled dataset with a 50% fraud rate positioned parallel to the production system (position A) having used only transactional features.

*4.3. Performance Measures*

We implemented all performance measures described in [4] (sensitivity (recall), specificity, precision, F-measure, G-mean, Mathew's correlation coefficient, balanced classification rate, average precision, and weighted average precision), finally settling on a specific family of measures with their accompanying charts:

**average precision** [65], defined as:

$$AP = \frac{\sum_{k=1}^{n}(P(k) * rel(k))}{\text{number of fraud transactions}} \tag{1}$$

A similar measure is defined in [66], which yields very similar results but is somewhat stricter when punishing erroneous predictions. The rationale behind both of these measures is the same and can be summed up as "algorithm A is superior to algorithm B only if it detects the frauds before algorithm B" [66]. We also computed a **weighted** version of the average precision where transactions were weighted according to the corresponding amount in €. Weighted measures were related to the non-trivial questions like "is it better to predict/prevent a single fraudulent transaction of 100€ or multiple transactions of 5€ each?". AP was chosen over all the other measures because it reflects (in one number) the ranking nature of this problem.

As we considered this problem a detection problem (and not "only" classification), we also used ranked precision and recall charts, both weighted and non-weighted, and AP aligned well with these charts in our experiments. Typical precision and recall charts are shown in Figure 5.

The abscissa shows the rank, which is the number of transactions ordered by fraud probability, and the ordinate shows the actual fraud percentage.

This kind of visualization maps well to the business case: a fraud detection system delivers transactions in the descending order of fraud probability. Human experts can examine only a portion of them—a number that certainly depends on the number of experts. At a certain point, the cost of (additional) experts exceeds the fraud losses. In any case, it is essential to achieve high precision at small ranks, and a sudden drop in precision could indicate a cut-off point in expert team size.

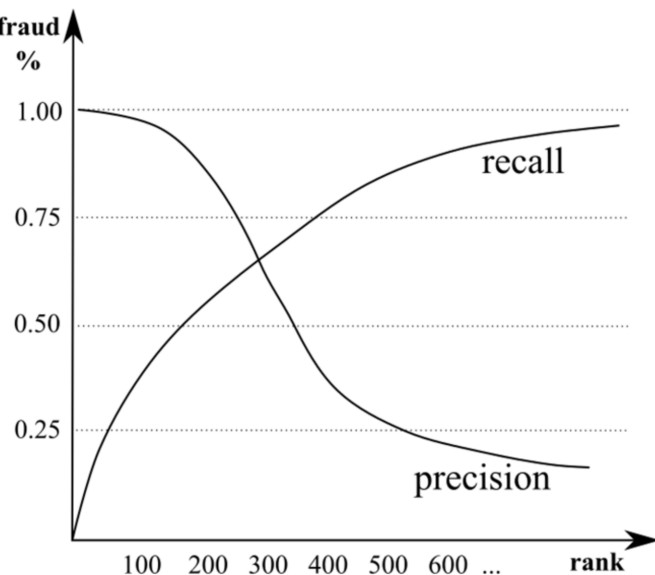

**Figure 5.** Precision and recall charts with characteristic curves.

## 5. Methodology

Our strategy to handle the proposed methodology was to start with a broader range of models and first determine the most promising ones, which we then analyzed in more depth. To this end, we first performed a set of preliminary experiments, following the procedure described below.

In the first set of experiments, we aimed to evaluate a broader range of models to determine the most promising ones. The models we considered in this preliminary experiment are as follows:

- Logistic regression (LR)—we used the L1 regularization and considered the regularization constant as a hyperparameter. This linear model is similar to the SM model.
- Multilayer perceptron (MLP)—is a fully connected neural network with one hidden layer. This model's advantage over the LR model is that it produces a nonlinear mapping from inputs to outputs. Thus, it may be able to better capture more complex interactions between input variables, which could lead to more accurate predictions. However, this model's nonlinear nature makes it much more prone to overfitting, which might offset the mentioned advantages. We used minibatch backpropagation to train the model. For regularization, we used dropout [67] and experimented with different numbers of neurons in the hidden layer (see Table 2 for details).
- Random forest (RF)—is an ensemble of decision trees learned on different feature and data subsets. This model is nonlinear and relatively robust to overfitting. This model's additional advantages are its short training time and a degree of interpretability of model decisions. Relevant hyperparameters were the minimal size of a tree node and the number of variables to possibly split at each node.

**Table 2.** Machine learning models and hyperparameters.

| Classifier | Hyperparameter | Values |
|:---:|:---:|:---:|
| LR | cost | 0.001, 0.001, 0.01, 0.1, 1, 100 |
| LR | regularization term in the loss | L1, L2 |
| MLP | number of neurons | 10, 20, 30 |
| RF | number of trees | 15, 20, 25 |
| RF | minimal node size | 10, 15, 20 |

To ensure the models will not overfit, we tuned the models' hyperparameters using three-fold cross-validation on the training set. Specifically, we ran three-fold cross-

validation on the training set for each hyperparameter combination and selected the best performing combination. We then trained the model on the entire training set using the best hyperparameter combination, labeled the test set, and obtained the final evaluation scores. A list of hyperparameters considered for each model and the corresponding values we experimented with are presented in Table 2.

The rest of this section describes a preliminary experiment list to optimize the models for these data. We report on AP, recall, precision, and F1 score, but note that the comparison results were similar for the other performance measures.

### 5.1. Scaling Input Data

First, we considered whether scaling the models' input features was beneficial to the performance. Scaling was done for each feature independently by standardizing its values across training examples.

Preliminary experiments showed that the MLP model without scaling performed very poorly and took a very long time to train, resulting in its omission from this experiment. For the other two models (LR and RF), we performed experiments on both scaled and unscaled versions of the data. Results are given in Table 3.

**Table 3.** Classifier performance with and without scaling.

| Classifier | Recall | Precision | F1 | AP |
|---|---|---|---|---|
| Baseline (RE) | 0.564 | 0.379 | 0.453 | 0.189 |
| LR without scaling | 0.806 | 0.255 | 0.387 | 0.472 |
| LR with scaling | 0.802 | 0.255 | 0.386 | 0.473 |
| RF without scaling | 0.835 | 0.370 | 0.514 | 0.697 |
| RF with scaling | 0.840 | 0.367 | 0.512 | 0.695 |

Scaling was slightly beneficial for the LR model while it slightly decreased the RF model's performance. Overall, the differences were tiny. In line with these findings in all subsequent experiments, we used the scaled version of the data with the LR and MLP models and the unscaled version with the RF model.

### 5.2. Feature Selection

In this experiment, we considered whether the results could be improved by ignoring some of the features that were not particularly predictive and effectively act as noise in the data. A simple and popular way of achieving this is $\chi 2$ feature selection, which uses a statistical test to assign a weight to each feature. Features that are better predictors get higher weights. We performed this experiment for linear (LR) and nonlinear (RF) model representatives and observed performance when using (a) only the 30 best features, (b) only the 100 best features, and (c) all available features (for brevity, we report here only these three representative categories, but experimented with a various number of features). In this experiment, we considered only the transaction features. Results are given in Table 4.

**Table 4.** Classifier performance for different feature count.

| Classifier | Recall | Precision | F1 | AP |
|---|---|---|---|---|
| Baseline (RE) | 0.564 | 0.379 | 0.453 | 0.189 |
| LR with 30 features | 0.818 | 0.205 | 0.327 | 0.311 |
| LR with 100 features | 0.813 | 0.230 | 0.359 | 0.429 |
| LR with all features | 0.803 | 0.254 | 0.586 | 0.472 |
| RF with 30 features | 0.846 | 0.338 | 0.483 | 0.709 |
| RF with 100 features | 0.853 | 0.357 | 0.504 | 0.725 |
| RF with all features | 0.836 | 0.371 | 0.514 | 0.696 |

Feature selection does not seem to affect the quality of results profoundly. For the LR model, the best results were achieved when using all features, while the RF model was

slightly better when using only the best 100 features. Taking these findings into account, in the rest of the experiment, we used the best 100 features for the nonlinear model (RF) and all features for the linear models (LR).

## 5.3. Classifiers Comparison

Finally, we now evaluate and compare the LR, MLP, and RF classifiers using described data scaling and feature selection properties. After much experimentation, we determined that the relative differences in model performance were entirely consistent across different subsampling types (5, 50, small, balance) and different model positions in the pipeline (A, B, or C). Therefore, we only report the results for the 5.sm subsampling method for the sake of brevity. The results in Table 5 represent the trends present across all subsampling methods and model positions.

**Table 5.** Classifier performance for 5.sm dataset and different model positions in the pipeline.

| Classifier | Recall | Precision | F1 | AP |
|---|---|---|---|---|
| Baseline (RE) | 0.564 | 0.378 | 0.453 | 0.189 |
| LR.5.A.sm | 0.454 | 0.766 | 0.570 | 0.388 |
| LR.5.B.sm | 0.454 | 0.763 | 0.569 | 0.387 |
| LR.5.C.sm | 0.451 | 0.759 | 0.566 | 0.385 |
| MLP.5.A.sm | 0.454 | 0.786 | 0.576 | 0.393 |
| MLP.5.B.sm | 0.441 | 0.707 | 0.543 | 0.354 |
| MLP.5.C.sm | 0.464 | 0.739 | 0.570 | 0.391 |
| RF.5.A.sm | 0.517 | 0.932 | 0.665 | 0.512 |
| RF.5.B.sm | 0.517 | 0.932 | 0.665 | 0.512 |
| RF.5.C.sm | 0.519 | 0.934 | 0.667 | 0.515 |

The best model across experiments was the RF model, with both LR and MLP performing somewhat worse. Interestingly, the baseline outperformed all models in terms of recall, but at the cost of much lower precision. Consequently, we limited our focus on the next section to the RF model and performed a more in-depth analysis.

## 5.4. Richer Features vs. More Complex Models

This section explores how particular aspects of the developed new models contribute to improved performance. Specifically, we addressed the following research questions:

1. How large is the difference between the baseline RE model and the developed models?
2. How much performance is gained by switching from a linear to a nonlinear model?
3. How much performance can be gained by including aggregated features in addition to trans features?

To this end, we tested the models in the best performing scenario—the C model position and 5.sm fraud (sampling) rate. We tested LR as a linear model and RF as the best nonlinear model and the RE baseline model. For the LR and RF models, we tested two versions of each, one using only trans features and one using both trans and additional aggregated features. Results are given in Table 6.

**Table 6.** Classifier performance for the 5.sm.C dataset and different feature sets.

| Classifier | Recall | Precision | F1 | AP |
|---|---|---|---|---|
| Baseline (RE) | 0.564 | 0.378 | 0.453 | 0.189 |
| LR.5.C.sm | 0.451 | 0.759 | 0.566 | 0.385 |
| LR.5.C.sm.trans | 0.375 | 0.749 | 0.500 | 0.308 |
| RF.5.C.sm | 0.519 | 0.934 | 0.667 | 0.515 |
| RF.5.C.sm.trans | 0.411 | 0.893 | 0.563 | 0.399 |

Concerning the first question, the results suggest that the newly developed models considerably outperformed the RE model on most metrics. To be fair, the new models are

optimized for the evaluation measures that we used in the evaluation, while the RE model is not, and these differences are somewhat expected. As for the second question, in most cases, the RF model was better, which is expected as it is nonlinear.

To answer the third question, we compared LR and RF variants using the aggregated features with those that do not use them. For both models, adding the aggregated features led to a significant performance improvement.

Finally, we can conclude that both (1) making the model nonlinear as well as (2) adding aggregated features both help improve performance. In this problem and dataset, the gains from using the nonlinear model were similar to gains from adding the aggregated features: both positively impacted the AP and F1 scores (Table 6). Moreover, using both modifications increased the performance even further, implying that their effects are complementary.

## 6. On Aggregated Features and Weighted Measures

Having decided on a random forest model with a 5% undersampling and "C" position, we now focus on a more in-depth comparison with the baseline model and comment and aggregated features and weighted measures. Although we only present "C" models here, everything stated also pertains to "A" models, despite them being marginally less effective. With that in mind, the following paragraphs will focus on models RF.5.C and RF.5.C.trans, that is, models with and without aggregated features. These will be compared with two existing models, ScoreModule and RuleEngine (Figure 4) using ranked precision and recall charts (Figure 5) in their basic and weighted variant, the latter using the monetary value of a transaction as a weight. We think that these charts align well with the business model since the processing of possibly fraudulent transactions has a limit on how many transactions can be processed in a specific unit of time by human analysts, so the ranking of possible frauds takes priority over other approaches of model evaluation.

Figures 6 and 7 show the behavior of precision at rank measure.

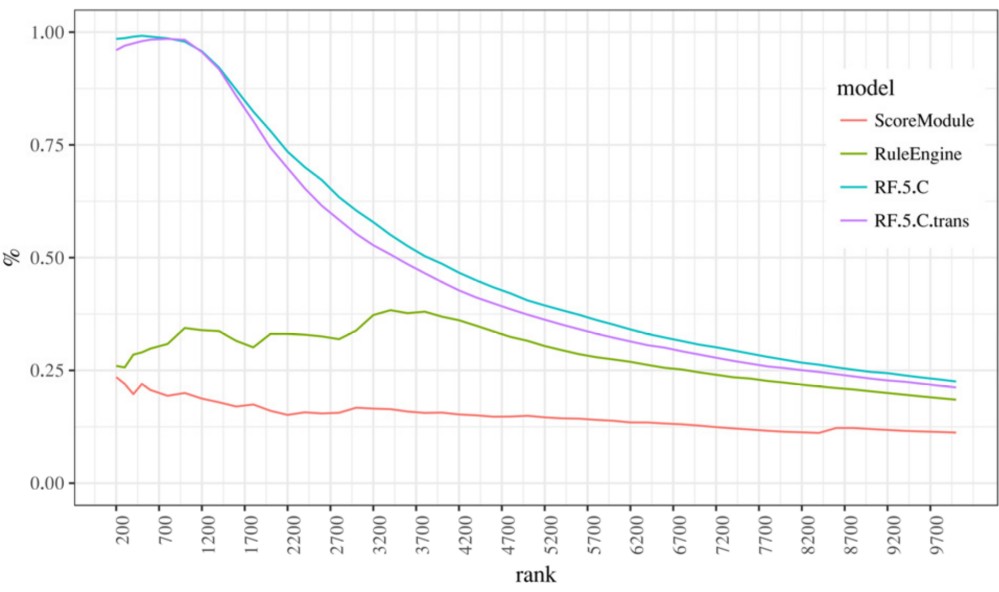

**Figure 6.** Ranked precision chart for the RF.5.C and baseline models.

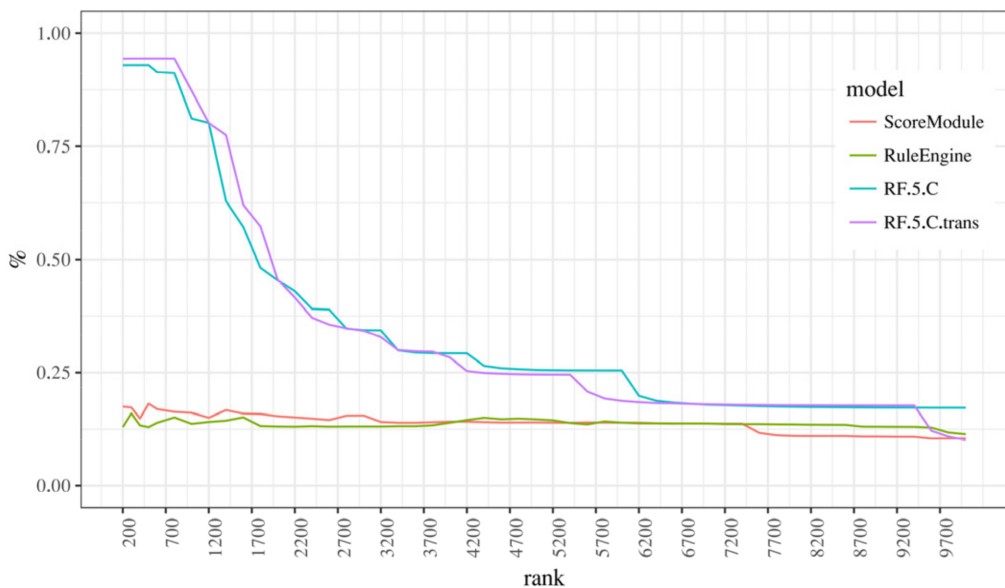

**Figure 7.** Weighted ranked precision chart for the RF.5.C and baseline models.

Succinctly put, it shows the "concentration" of fraudulent transactions at a certain rank when they are ordered by the fraud probability. It is apparent that random forest models significantly outperform existing models in both regular and weighted variants. This means that when the new models are very confident something is a fraud, we can tell with a high degree of certainty that the transaction is, in fact, fraudulent, and if we analyze transactions by rank, we can expect to see a very high concentration of fraudulent transactions at the top ranks (nearly 100%) compared to existing models where the concentration oscillates between 25% and 40% (for RuleEngine), or starts at 25% and slowly declines (ScoreModule).

Figures 8 and 9 depict recall at rank measure, weighted, and basic variant. The recall measure states the ratio of frauds in the subset of transactions at the current rank compared to the entire dataset.

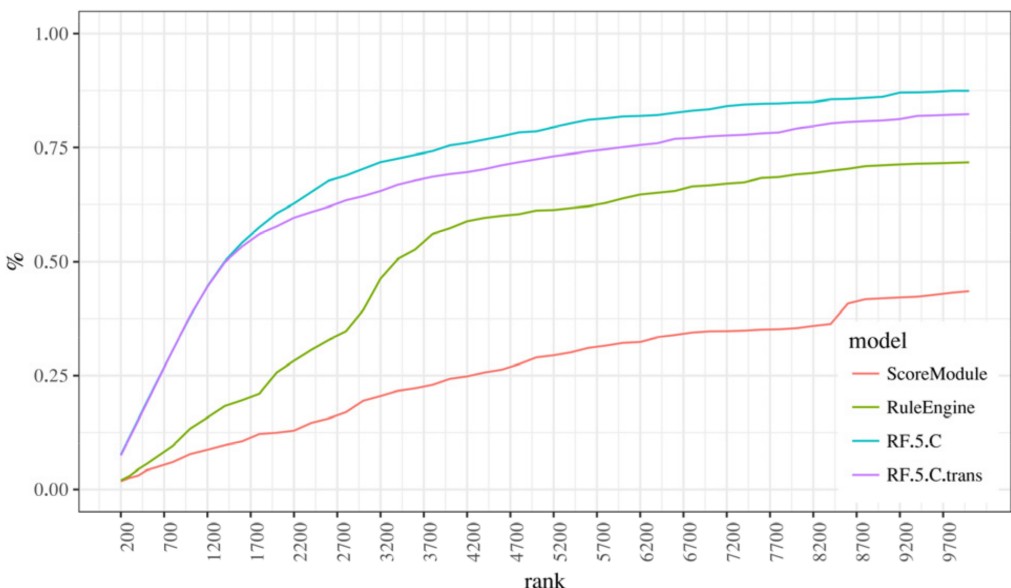

**Figure 8.** Ranked recall chart for the RF.5.C and baseline models.

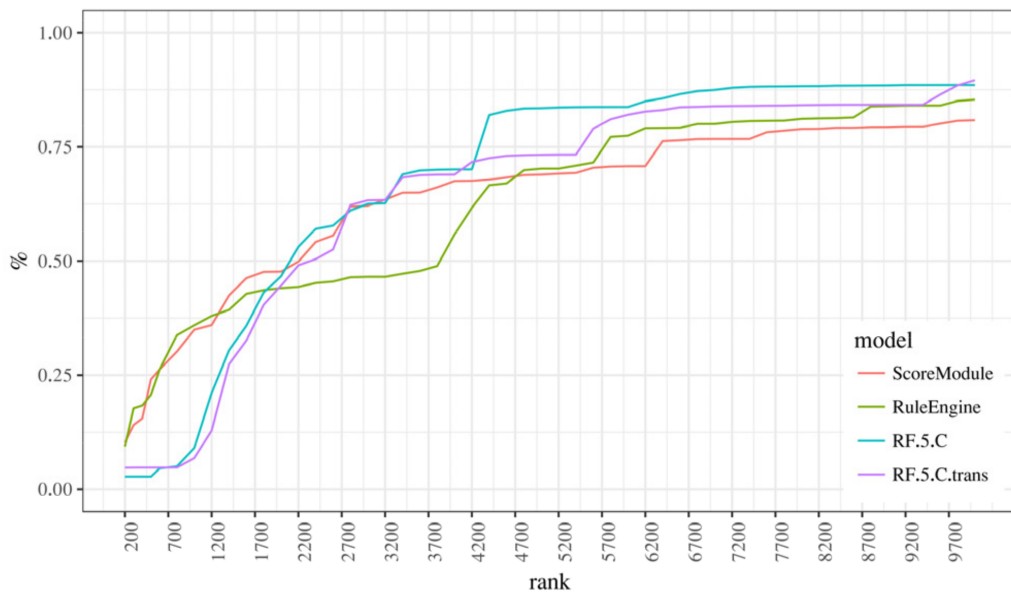

**Figure 9.** Ranked weighted recall chart for the RF.5.C and baseline models.

As can be seen, Figure 8 confirms that new models outperformed the existing ones, in other words, that the concentration of fraud is so high at the top ranks that the first 1000 transactions (which is 1.67% of the total number of around 60,000 transactions in the test dataset) "catch" 30% of frauds from the entire dataset. Figure 9, which shows the weighted recall graph is more interesting: contrary to the first three graphs, RF.5.C models underperformed compared to the ScoreModule and RuleEngine models until rank 1700 (RF.5.C) or rank 3200 (RF.5.C.trans), where the random forest models started to take over.

This occurs because weighted recall states how many frauds are "caught" at a certain rank compared to the total number of frauds, with the monetary amount of the fraud being used as a weight. Weighted measure rewards identifying frauds with higher monetary values more, with the amount of "reward" proportional to the monetary amount. In practice, however, most frauds involve small or very small amounts of money, which has an interesting implication—the amount of money involved in a (fraudulent) transaction has a lot of predictive power when deciding whether the transaction is fraudulent or not.

Therefore, when a predictive model tasked with discovering patterns related to fraud is asked to rank transactions based on the probability of fraud, the frauds with smaller amounts will naturally (and rightfully) be ranked above those with higher amounts.

Figure 10 clearly depicts this relationship: note the average amounts for transactions equal to and over 85% fraud probability rate, which were all below the total average.

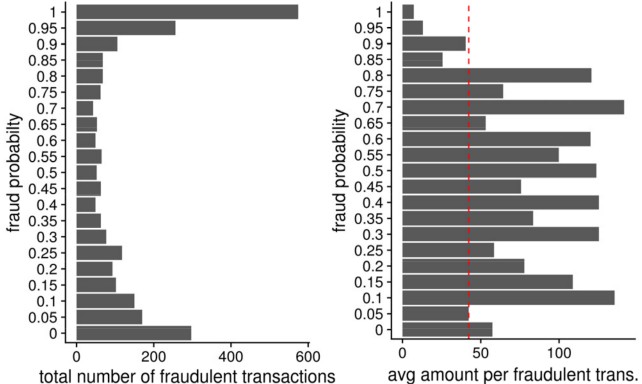

**Figure 10.** Comparing the total number and average amount of fraudulent transactions.

This behavior raises interesting questions when deciding on a direction to take when building future predictive models. Instinctively, we might decide to construct models that prioritize frauds with a higher monetary value, compromising prediction accuracy and letting more small amount of frauds "slide" instead of building models that focus on better fraud prediction of what is and is not a fraud. Such monetary-oriented models would not be necessarily better from a business perspective because individual frauds do not exist in a vacuum. There are additional patterns in fraudulent behavior that can have a much more intricate and complex effect on actual damage to the business. For example, there is a typical fraud pattern where frauds with larger amounts are preceded with "testing" frauds using smaller amounts, which in turn means that timely identification of fraud with a small amount might have a much larger "weight" when it comes to evaluating the possible impact on business than what the weighted recall measure might bestow it. Ultimately, this all means that building efficient predictive models regarding business needs requires a cautious and deliberate approach that must address a lot of individual factors besides the accuracy of the model or the immediate monetary impact. While machine learning models can be very effective in identifying ranking frauds, the final decision on how to use the output of these models (and evaluate their performance) must be first and foremost made from a business perspective. Additionally, we do not see the role of a human analyst at the end of the fraud detection pipeline diminishing anytime soon—the goal is not to downplay their role but to provide them with better tools and timely information.

## 7. Discussion

In this section, we comment on the challenges outlined in Section 2 in the context of our use-case and data. Scalability was not addressed directly as it is domain and technology-specific. However, as a scalability side note, the A, B, and C models' similar performance favors A models as a more independent and thus more scalable solution. A model can be employed in a horizontally scalable fashion with multiple identical models working in parallel. Concept drift was also not examined here, but it should be noted that the authors in [66,68] presented an overview of various windowing strategies to deal with this problem.

Our research shows that significant gains can be achieved by investing in feature engineering, which is not surprising and agrees with the literature (e.g., [42]). Additional work should be done to reduce the feature set cardinality without significant performance drops, as this would speed up the processing and allow for additional models that do not perform well with large feature sets. In our experiments, RF even performed slightly better with a reduced set of features (100 vs. 300 features). In addition, it would be interesting to try and convert the rules from the rule engine (at least the most useful ones) to features and thus tap into the accumulated domain knowledge. Some of them probably already are. For instance, the rule "small transaction followed by a large transaction" is reflected in aggregated features, for example, "the number of transactions in the last 10 min" and "last transaction amount" and, of course, the transaction amount. Cross-referencing these sets is not an easy task, but we believe it is worth doing.

Undersampling is the most common approach to balancing class sizes, and it is reported to work well [4]. Our research suggests that it is not advisable for a company with limited resources to invest in experimenting with different sampling methods (over/undersampling, hybrid, filtering, ensemble methods, etc.). Our experiments did not show a significant difference in additionally undersampled sets as the 50% sets did not perform significantly differently from the 5% set. This is favorable as the 50% set is an order of magnitude smaller and is easier to handle, train, and update models, positively impacting scalability in general.

Many performance measures can cloud the comparison, and we have proposed a single numerical measure: average precision with its weighted counterpart. Additionally, we found that ranked precision and recall charts are very informative and map well to the business case as this is a ranking rather than a classification problem. An interesting issue

arose in our experiments—weighted measures (i.e., models behave somewhat differently than their non-weighted versions, especially when the weighted recall is considered). The importance of the amount is not a trivial question and should be discussed with the business partner. There is also a non-obvious technical detail hidden here: the weighted measure can be somewhat manipulated. Specific algorithms (e.g., RF) generate very coarse probability resolutions and produce many transactions with the same probability (e.g., ten transactions with 0.75 fraud probability). A secondary descending sort by amount can significantly impact the weighted measure. On the other side of the spectrum, an algorithm can produce a too finely grained probability (e.g., 0.9123 and 0.9122), where such precision is meaningless. In such cases, transactions could be binned (e.g., bin "0.91") and then additionally sorted by amount. Then again, how coarse a bin to choose?

When it comes to algorithm selection, in our research, the random forest algorithm performed best, which is consistent with the scientific literature where RF, or its variants, is the most often mentioned and recommended algorithm [4]. Therefore, we deem the enrichment of the existing fraud detection system with the random forest algorithm the key component that improves the overall performance. It is recommended to focus on the RF, and try to evaluate the best set of hyper-parameters (e.g., the number of trees in the forest), and potentially explore certain modifications to that model (for instance, in [69] a "balanced RF" algorithm was used, which is a modification of the classic random forest). In addition to RF, the logistic regression and the MLP algorithms were evaluated. All models showed a certain improvement over the baseline models, but RF proved to be the best. Scaling features had no dramatic impact on RF and LR, while for MLP, it was crucial to obtain acceptable results. Model C appears to be consistently only slightly better than A and B, which suggests that model position is not crucial and that the model can learn what SM and RE know. Position independence is good news because it leaves more freedom in designing the new system architecture (and allows less or no coupling of the new model with the existing production system).

In the conclusion of this section, we will outline our general guidelines for adding machine learning support to existing fraud detection system. First, we will assume that this existing system relies on data collected from previous transactions and a collection of rules that were devised from domain expertise and exploratory analysis of this data. The next step is collecting and cleaning the data to remove any inconsistencies, and optionally performing some feature engineering in the form of removing unnecessary columns containing redundant information as well as adding columns using domain expert rules as guides as to what information might be considered predictive when it comes to fraud detection. The data then needs to be stored, with a focus on storing as much data as possible including the most current as well as historical data while having availability and latency in mind. In the case of large data volumes, big data solutions such as Apache Hive might be considered; otherwise, a classic relational database (perhaps with a cache layer using e.g., Redis) should be a preferred choice.

Then, two machine learning modules should be introduced: a training module able to create a random forest model and a classifier module that would implement this model and then be integrated directly in the fraud detection system, assigning fraud probability values to incoming transactions in near-real time. When choosing a training dataset, we recommend a sampling technique that would collect as large a dataset as possible considering available resources and specifics of the machine learning platform, while favoring the more current data as well as taking fraudulent transactions over non-fraudulent ones, achieving a balanced ratio between them. After making the initial random forest model, the training module should be periodically used to replace older models with newer ones that would be able to detect newer fraud patterns. The retention rate should be a business decision, possibly dictated in part by the estimated performance of the currently deployed model. Another business decision would dictate how the random forest probabilities would be used when it comes to actually addressing frauds—these could be used simply as an additional flag complementing the existing system and information it provides, or

they could influence the order the possibly fraudulent transactions are forwarded to the experts, prioritizing those transactions that are most likely fraudulent or—as discussed previously—those that are both likely to be fraudulent but also have other characteristics that negatively impact the business such as containing large monetary amounts.

## 8. Conclusions

This paper has researched how to cost-efficiently enhance a real-world credit card fraud detection system with data mining models. We identified major challenges in this area: feature engineering, scalability, unbalanced data, concept drift, performance measures, and model algorithm selection. The research shows room for improvement in the existing system and that one should foremost invest in feature engineering and model tuning. All data mining models performed better than the existing system, whereas random forest performed best. We empirically confirmed many of the literature findings and detected an interesting, weighted measure aspect of fraud detection, which presents further research. We proposed apt performance measures for model validation—average precision and ranked precision/recall charts as we see this as the ranking, and not a binary classification task. A carefully designed set of aggregated features, which can be viewed as a card/user profile, makes a difference, and rule engine rules containing precious domain knowledge should also be considered in its construction. As for (under)sampling and concept drift, we recommend using the already developed state-of-the-art solutions and not investing further in custom solutions in this area, at least not initially. Our insights were obtained on a very large dataset that is representative of credit card fraud including collaboration with domain experts. Consequently, we believe that the insights are relevant and generalize well to similar datasets of other credit-card companies as well as related types of fraud.

**Author Contributions:** Conceptualization and methodology, I.M., M.K., D.P. and L.B.; software, visualization and analysis, I.M., M.K. and D.P.; investigation, L.B.; writing—original draft preparation, I.M., M.K., D.P. and L.B.; writing—review and editing, I.M. and L.B.; project administration, I.M.; funding acquisition, I.M. All authors have read and agreed to the published version of the manuscript.

**Funding:** This research was funded by the European Regional Development Fund under the grant KK.01.1.1.01.0009 (DATACROSS).

**Institutional Review Board Statement:** Not applicable.

**Informed Consent Statement:** Not applicable.

**Data Availability Statement:** Not applicable.

**Acknowledgments:** The authors would like to thank Bojana Dalbelo-Bašić for her valuable advice and insights that helped improve this paper.

**Conflicts of Interest:** The authors declare no conflict of interest.

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
