# Peer review of "Credit Card Fraud Detection in Card-Not-Present Transactions: Where to Invest?"

_applsci, doi:10.3390/app11156766_

Round 1

Reviewer 1 Report

  1. "E-commerce is founded on credit cards as means of payment and credit card adoption" (line 26) - The expression look to be exclusive therefore, I recommend reformulation
  2. "A significant number of papers have been published in the last ten years on the subject" (line 59) - There is no review of the literature in this area, therefore I recommend completing and mentioning some important works specific on applied scope / field (economic, legal, operational, etc.). As an example, can be mentioned „Găbudeanu, Larisa, Iulia Brici, CodruÈ›a Mare, Ioan Cosmin Mihai, and Mircea Constantin Șcheau..2021. Privacy Intrusiveness in Financial-Banking Fraud Detection. Risks 9: 104. https://doi.org/10.3390/risks9060104”
  3. "This issue can be addressed on an algorithmic level by adjusting the data mining algorithm but is typically addressed on the data" (line 108) – How are addressed? I recommend a very...very short clear presentation of differences (one or maximum 2 phrases for introduction of the rest of the article)

  4. The figures differ as a way of realization depending on the source (ex: Figures 1 and 2 are different from Figure 3 etc.) - I recommend the uniformity (as much as possible)

Author Response

In short, all objections have been addressed:

  1. "E-commerce is founded on credit cards as means of payment and credit card adoption" (line 26) - The expression look to be exclusive therefore, I recommend reformulation
    • Response: reformulated to: E-commerce heavily relies on credit cards as means of payment
  2. "A significant number of papers have been published in the last ten years on the subject" (line 59) - There is no review of the literature in this area, therefore I recommend completing and mentioning some important works specific on applied scope / field (economic, legal, operational, etc.). As an example, can be mentioned „Găbudeanu, Larisa, Iulia Brici, CodruÈ›a Mare, Ioan Cosmin Mihai, and Mircea Constantin Șcheau..2021. Privacy Intrusiveness in Financial-Banking Fraud Detection. Risks 9: 104. https://doi.org/10.3390/risks9060104”
    • Response: reformulated, added reference to the aforementioned paper, and also to another systematic review (https://link.springer.com/chapter/10.1007/978-3-030-38501-9_29 )
  3. "This issue can be addressed on an algorithmic level by adjusting the data mining algorithm but is typically addressed on the data" (line 108) – How are addressed? I recommend a very...very short clear presentation of differences (one or maximum 2 phrases for introduction of the rest of the article)
    • Response: reformulated, added a very short explanation and a reference for those who want to know more: "This issue can be addressed on an algorithmic level but is typically addressed on the data level [19]. At the algorithmic level, algorithms themselves are adjusted to cope with the detection of the minority class, while at the data level a pre-processing step is performed to rebalance the dataset. Various pre-processing techniques…"
  4. The figures differ as a way of realization depending on the source (ex: Figures 1 and 2 are different from Figure 3 etc.) - I recommend the uniformity (as much as possible)
    • Response: Fig1 and fig2 are outsources (referenced) and that is why they stand out. However, we have recreated the fig1 using R and ggplot2 to increase uniformity. Fig2 does not have explicit/exact numbers for series (data points) so we could not retype them in R and recreate fig2, so fig2 is left as is and is the only figure that somewhat stands out.

Reviewer 2 Report

This paper evaluated different classifiers for the best suitability of predicting fraud transactions in Card-Not-Present user cases. This is more of an experimental paper rather than theoretical research. The model selecting approach is a two-phase process: the authors start with a broad range of models including LR, MLP, and RF and then further focus on RF after finding out its average performance dominates. 

This paper investigates different models based on practical industrial data, which brings a great deal of significance to the study. I would like to recommend the paper to be published after the presentation is improved. My comments are as follows.

1. The figure at Page 11 has no caption. The caption might have gone to Page 12. This figure also lacks unit on horizontal axis and label for the vertical axis is to be explained in detail. 

2. Fig. 7 is at poor quality and has the same issue as Fig. 6. Please regenerate this figure and improve the presentation. 

3. Fig. 8 & 9 are in poor quality and should be improved in the same way as Fig. 6 & 7. 

4. Fig 10 has no tics on horizontal axis.

5. Table 4, caption and the table are located in separate pages. Please shift to the same. 

6. Line 354, "Chi-square" symbol was not well posed. 

7. The symbolic system - "RF.50.A.sm.trans" and others alike are too confusing to follow. Please change to a clearer annotation. 

Author Response

In short, all objections have been addressed:

  1. The figure at Page 11 has no caption. The caption might have gone to Page 12. This figure also lacks unit on horizontal axis and label for the vertical axis is to be explained in detail.
  • Response: We would like to thank the reviewer for pointing out these issues with figures and tables. We have improved the quality for all figures (from 6 to 10) and connected them with their captions (there was an issue with Word’s default image compression settings). Figure 6 on page 11 now has rank on horizontal axis (rank is just an integer with no unit), and percentage on vertical axis. Figures 7, 8 and 9 also have a percentage on the vertical axis. Figure 10 now also features a red dashed line marking the total average (mentioned in the text).
  1. Fig. 7 is at poor quality and has the same issue as Fig. 6. Please regenerate this figure and improve the presentation. 
  • Response: The quality for the figure is improved. The figure caption is now next to the figure.
  1. Fig. 8 & 9 are in poor quality and should be improved in the same way as Fig. 6 & 7. 
  • Response: The quality for the figures is improved.
  1. Fig 10 has no tics on horizontal axis
  • Response: The improved version has ticks on horizontal axis. Also, Fig10 now also features a red dashed line marking the total average (mentioned in the text).
  1. Table 4, caption and the table are located in separate pages. Please shift to the same. 
  • Response: The table 4 caption is now next to the table.
  1. Line 354, "Chi-square" symbol was not well posed. 
  • Response: We tried to fix this but in word it can't be better than it is now. We put „chi-square“ in equation but the difference is slight.
  1. The symbolic system - "RF.50.A.sm.trans" and others alike are too confusing to follow. Please change to a clearer annotation. 
  • Response: we would gladly change to a clearer notation but each of these five positions in the notation stands for a different parameter. In an attempt to be clearer, we have introduced the convention of leaving out last two parameters which then defaults to integral data set and full feature set. That way, none of the figures or tables uses the fully expanded form, e.g. figures 6-9 use RF.5.C and RF.5.C.trans.